# Distress and Sensitization as Main Mediators of Severity in Women with Fibromyalgia: A Structural Equation Model

**DOI:** 10.3390/biomedicines10051188

**Published:** 2022-05-20

**Authors:** Bernard X. W. Liew, Juan Antonio Valera-Calero, Umut Varol, Jo Nijs, Lars Arendt-Nielsen, Gustavo Plaza-Manzano, César Fernández-de-las-Peñas

**Affiliations:** 1School of Sport, Rehabilitation and Exercise Sciences, University of Essex, Colchester CO4 3SQ, UK; liew_xwb@hotmail.com; 2Department of Physiotherapy, Faculty of Health, Universidad Camilo José Cela, Villanueva de la Cañada, 28692 Madrid, Spain; 3VALTRADOFI Research Group, Department of Physiotherapy, Faculty of Health, Universidad Camilo José Cela, Villanueva de la Cañada, 28692 Madrid, Spain; 1umutvarol7@gmail.com; 4Pain in Motion Research Group (PAIN), Department of Physiotherapy, Human Physiology and Anatomy, Faculty of Physical Education & Physiotherapy, Vrije Universiteit Brussel, 1050 Brussels, Belgium; jo.nijs@vub.be; 5Department of Physical Medicine and Physiotherapy, University Hospital Brussels, 1050 Brussels, Belgium; 6Department of Health and Rehabilitation, Unit of Physiotherapy, Institute of Neuroscience and Physiology, Sahlgrenska Academy, University of Gothenburg, 40530 Gothenburg, Sweden; 7Center for Neuroplasticity and Pain (CNAP), SMI, Department of Health Science and Technology, Faculty of Medicine, Aalborg University, 9220 Aalborg, Denmark; lan@hst.aau.dk; 8Department of Medical Gastroenterology, Mech-Sense, Aalborg University Hospital, 9000 Aalborg, Denmark; 9Department of Radiology, Rehabilitation and Physiotherapy, Universidad Complutense de Madrid, 28040 Madrid, Spain; gusplaza@umc.es; 10Instituto de Investigación Sanitaria San Carlos (IdISSC), 28040 Madrid, Spain; 11Department of Physical Therapy, Occupational Therapy, Rehabilitation and Physical Medicine, Universidad Rey Juan Carlos, 28922 Alcorcón, Spain

**Keywords:** fibromyalgia, structural equation modelling, sensitivity, distress, severity

## Abstract

We aimed to explore a path model identified using a structural equation model (SEM) which best explains the multivariate contributions of sensitization, sensitivity, and emotional variables to clinical severity in women with FMS. Pain features, the Central Sensitization Inventory (CSI), painDETECT, S-LANSS, the Hospital Anxiety and Depression Scale (HADS), the Pittsburgh Sleep Quality Index (PSQI), the Pain Catastrophizing Scale (PCS), the Pain Vigilance and Awareness Questionnaire (PVAQ), the 11-item Tampa Scale for Kinesiophobia (TSK-11), and pressure pain thresholds (PPTs) were collected from 113 women with FMS. Four latent variables were created: severity (clinical pain features), sensitivity (PPTs), sensitization (S-LANSS, CSI, painDETECT), and distress (HADS-A, HADS-D, PCS, PVAQ, TSK-11). Data fit for the measurement model were considered excellent (RMSEA = 0.043, CFI = 0.966, SRMR = 0.067, and NNFI = 0.960). Distress had a significant relationship with the mediators of sleep (*β* = 0.452, *p* = 0.031) and sensitization (*β* = 0.618, *p* = 0.001). The only mediator with a significant effect (*β* = 1.113, *p* < 0.001) on severity was sensitization. A significant indirect effect of sensitization (*β* = 0.687, *p* = 0.001) that explained the relationship between distress and severity was also identified. The proposed model suggests that distress and sensitization, together with poor sleep, have a complex mediating effect on severity in women with FMS. The identified path model can be leveraged in clinical trials investigating treatment approaches for FMS.

## 1. Introduction

Fibromyalgia syndrome (FMS) is a pain condition with a worldwide prevalence of 6.6% [1]. A recent review reported an overall median incidence of 4.3 per 1000 person-years in the general population [2]. Its symptomatology is highly heterogeneous and includes widespread pain, fatigue, stiffness, exacerbated responses to different stimuli, sleep disorders, mood disturbances, and cognitive dysfunction [3]. More recently, epigenetic modifications have indicated the importance of environmental stressors in fibromyalgia [4,5,6].

This plethora of manifestations suggests complex interactions explaining the heterogeneity in the clinical presentation observed in people with FMS. In fact, there is evidence suggesting the presence of different subgroups of women with FMS combining higher or lower sensitivity, and more or less stress [7,8,9,10]. These studies used pain- and disability-related, cognitive, or psychological aspects to identify subgroups of patients with FMS using cluster analysis. Additionally, considering that one of the most common features of FMS is hypersensitivity to pressure pain (expressed as a decreased pressure pain threshold), these previous studies did not include this psychophysiological outcome when evaluating altered nociceptive pain processing in their analyses [7,8,9,10]. The application of cluster analysis is able to identify subgroups of individuals based on common variables; however, it does not identify complex interactions or mediating effects between the variables.

Since an ideal theoretical framework of FMS integrates reciprocal interactions between biology (clinical and sensory aspects) and behaviors (psychological and cognitive aspects) [11], we argue that a structural equation modelling (SEM) approach can provide an efficient framework to test-explore-retest competing causal models for a complex disorder such as FMS. In fact, when quantifying complex multivariate mechanistic pathways are included, variables can simultaneously depend on, and influence, other variables. SEM is proposed as a “de facto” statistical method. A recent study used SEM to assess the relevance of psychological factors such as pain acceptance, catastrophizing, and coping strategies in pain severity in women with FMS [12]. However, this study only included psychological/cognitive variables [12]. Accordingly, the aim of this study is to explore and validate a path model identified using SEM that best explains the potential multivariate contributions of sensitization, sensitivity, and emotional variables to clinical severity in women with FMS.

## 2. Materials and Methods

### 2.1. Participants

Women with a medical diagnosis of FMS [13] were voluntarily recruited from the AFINSYFACRO Fibromyalgia Association, Madrid (Spain), using local announcements. Exclusion criteria included: (1) previous surgery; (2) diagnosis of a neuropathic pain condition (e.g., radiculopathy or myelopathy); (3) a comorbid underlying medical condition (e.g., tumor); (4) whiplash injury; (5) pregnancy; (6) use of medication affecting muscle tone or pain perception. This study was approved by the Local Ethics Committee of Camilo José Cela University (UCJC 20-10-2020) and Universidad Rey Juan Carlos (URJC 08-30-2020). Participants signed written informed consent forms prior to their inclusion in the study.

### 2.2. Severity Variables

Pain-related variables, e.g., pain intensity and pain extent, were considered as severity variables. The worst pain intensity at rest, as well as intensity of pain during daily living activities, were assessed on a 10-point Numerical Pain Rate Scale (NPRS, 0: no pain, 10: maximum pain) [14]. In addition, participants were instructed to complete a pain drawing using a red pencil to shade their pain symptoms on ventral and dorsal women’s body charts printed on an A4 sheet. Paper body charts were stored as pdf files and imported into an online platform (https://syp.spslab.ch, accessed on 23 June 2021) that uses a custom algorithm. Pain extent was expressed as a percentage of the total body chart’s area (ventral: 476,650 pixels, dorsal: 489,592 pixels). The reliability of this pain extent calculation procedure was shown to be high in patients with chronic pain [15].

### 2.3. Sensitization Variables

We assessed the presence of self-reported, sensitization-associated, and neuropathic pain symptoms. The Central Sensitization Inventory (CSI) is a 25-item self-reported questionnaire which assesses the presence of sensitization-associated symptoms [16]. Each item is evaluated on a 5-point Likert scale which results in a total score from 0 to 100 points. A cut-off value of 40 points suggests altered nociceptive pain processing.

The self-reported Leeds Assessment of Neuropathic Symptoms and Signs (S-LANSS) [17] and painDETECT [18] questionnaires were used to assess the presence of neuropathic pain symptomatology. The S-LANSS classifies patients into a predominantly or non-predominantly neuropathic pain component if the total score is ≥12 out of 24 points [16]. The painDETECT questionnaire (total score of 0–38 points) uses the following classification: unlikely neuropathic pain component (<12 points), ambiguous (12–18 points), or likely neuropathic component (>18 points) [18].

### 2.4. Sensitivity Variable

Pressure pain thresholds (PPTs) were bilaterally evaluated by a trained/experienced assessor who was blinded to the outcomes of the remaining variables. PPTs were assessed over the mastoid process, upper trapezius, elbow, hand, posterosuperior iliac spine, greater trochanter, knee, and tibialis anterior with an electronic algometer (Somedic AB©, Farsta, Sweden) to assess widespread sensitivity to pressure pain. The pressure was applied at a rate of approximately 30kPa/s on each point. The mean of 3 trials on each point, with a resting period of 30 s between each, was calculated [19]. The mean of both sides was used in the analysis.

### 2.5. Distress Variables

Different emotional and cognitive variables were used to evaluate distress. The Hospital Anxiety and Depression Scale (HADS) was used to evaluate anxiety levels (HADS-A, 7 items, 0–21 points) and depressive levels (HADS-D, 7 items, 0–21 points) [20]. The cut-off scores suggestive of clinical anxiety and depressive symptoms (HADS-A ≥ 12 points; HADS-D ≥ 10 points) were considered.

The short-form 9-item Pain Vigilance and Awareness Questionnaire (PVAQ) was used to evaluate pain hypervigilance, e.g., ideas about observing, monitoring, and focusing on pain [21]. The 11-item short-form of the Tampa Scale for Kinesiophobia (TSK-11, 0–44 points) was used to quantify the fear of movement experienced by patients [22]. Finally, the Pain Catastrophizing Scale (PCS, 0–52 points) was used to assess pain catastrophizing responses (e.g., rumination, magnification, and despair aspects) in individuals with pain [23].

### 2.6. Statistical Analysis

#### 2.6.1. Packages

All analyses were performed using R software (v4.0.2). The following packages were used: mice [24] for data imputation, lavaan [25] for SEM analysis, semPlot [26] for visualizing SEM paths, and semTools [27], which fits an SEM model across our 20 imputed datasets and pools the statistical outputs using Rubin’s rules. All codes and results are included in a public online repository (https://bernard-liew.github.io/2020_cts_bn/5-fms.html, accessed on 11 November 2021).

#### 2.6.2. Missing Data Management

Multiple imputations were performed on all variables with missing values using the multivariate imputation by chained equations method [24]. The random forest method was used for imputation. We generated 20 imputed datasets using a maximum number of 30 iterations for each imputation.

#### 2.6.3. Structural Equation Modelling (SEM)

All continuous variables are scaled to a mean of 0 and a standard deviation (SD) of 1 prior to modelling. SEM are probabilistic models that unite multiple predictors and outcome variables in a single model, and where latent variables can also be included. First, SEM was used to assess the fit of the proposed measurement model which defines the relationship between the observed variables and the latent variables of severity, sensitivity, sensitization, and distress (Figure 1). Next, SEM was used to fit the path model, which was informed by the current FMS literature (Figure 2). In simple mediation models, with only one mediator, the effect of the independent variable on the mediator is denoted by *a*, the effect of the mediator on the dependent variable is denoted by *b*, the direct effect of the independent variable on the dependent variable is denoted by *c*′, and the indirect effect of the mediator by the product is denoted by *a* × *b*, or simply *ab*. In a parallel mediation model with three mediators, there are three *a*, *b*, and *ab*.

For both the measurement and path models, maximum likelihood was used to estimate the model’s parameters, while the Huber–White robust standard errors were used. In the present study, we report the standardized model’s parameters, i.e., a 1 SD change in the independent variable predicts a 1 SD change in the dependent variable. An excellent model fit is determined when two of the four fit indices exceed the thresholds: (a root mean square error of approximation [RMSEA] ≤ 0.05; a standard root mean residual [SRMR] ≤ 0.05; a confirmatory fit index [CFI] ≥ 0.95; and a non-normed fit index [NNFI] ≥ 0.95) [28]. The 95% confidence interval (CI) of estimated parameters was estimated using Monte Carlo bootstrapping. For the estimated parameters, a *p*-value < 0.05 was considered to be statistically significant.

## 3. Results

Of the 127 women with FMS initially screened for eligibility criteria, 8 were excluded due to previous surgery, four had suffered a previous whiplash, and two were pregnant at the moment of the study. Finally, 113 women (mean age: 52.5 ± 11 years) were included. Table 1 shows the descriptive features of the cohort.

The tested measurement model and associated standardized regression weights are reported in Figure 1. Fit for the measurement model was excellent (RMSEA = 0.043, CFI = 0.966, SRMR = 0.067, and NNFI = 0.960). The model and associated standardized regression weights are reported in Figure 2. The standard errors, 95% confidence intervals (CI), and *p*-values can be found in Table 2. The model had fit values of RMSEA = 0.039, CFI = 0.970, SRMR = 0.066, and NNFI = 0.975, reflecting an adequate model fit.

There was a significant total effect of distress on sleep (*β* = 0.745, *p* < 0.001) (Figure 2, Table 2). Distress had a significant relationship with the mediators of sleep (*β* = 0.452, *p* = 0.031) and sensitization (*β* = 0.618, *p* = 0.001) (Figure 2, Table 2). The only mediator with a significant effect (*β* = 1.113, *p* < 0.001) on severity was sensitization (Figure 2, Table 2). There was a significant indirect effect of sensitization (*β* = 0.687, *p* = 0.001) that explained the relationship between distress and severity (Figure 2, Table 2).

## 4. Discussion

Current understanding supports the presence of biopsychosocial mechanisms associated with FMS, which are suited to analysis using an SEM framework. The relevance of psychological/cognitive factors in women with FMS was recently observed using an SEM framework [12]. This study applied SEM to validate a multivariate pathway model to better understand the complex interactions between psychological, neuro-physiological, and clinical variables in women with FMS. Latent variables were based on the most up-to-date literature supporting a heterogeneous and complex association between biological and psychological aspects of FMS (Figure 1) and more recent environmentally provoked factors [4]. Sleep quality was not included in any latent variable since it represents a different aspect not involved in the other latent variables. We found that the measurement (creation of latent variables) and path models were supported by the SEM.

The primary finding of the path model revealed that distress (i.e., psychological/emotional variables) and sensitization (i.e., self-reported outcomes) were factors mediating severity (i.e., clinical features) in women with FMS. The fact that stress [29] and psychological/emotional disorders [30] promote central sensitization, pain, and related disability in chronic pain conditions is supported by the current literature. Angarita-Osorio et al. recently observed that emotional and cognitive factors are associated with higher pain and related disability in women with FMS [31], supporting our results regarding the role of distress on severity. Similarly, other factors included in distress, such as kinesiophobia or catastrophism, also mediate the association between pain and sensitization in people with FMS [32,33]. Previous and current results support the relevant role of distress factors (emotional and cognitive aspects) in people with FMS, explaining the effectiveness of psychological and cognitive interventions in these patients [34].

Poor sleep quality was identified as another risk factor for developing widespread pain, since sleep deprivation impairs descending pain inhibition pathways [35]. Poor sleep can also promote fatigue by inducing a daily low-energy state leading to a prolonged tiredness sensation [36], which can then promote distress. Similarly, poor sleep/lack of sleep also activates pro-inflammatory pathways [37]. Furthermore, poor sleep impairs descending pain modulation, which may cause widespread sensitization [38]. Both mechanisms, distress and poor sleep, are involved in the pathogenesis of FMS. Our path model found that distress directly affected sleep quality in women with FMS. Accordingly, treatment strategies targeting sleep in women with FMS should be applied, as indicated for many other chronic pain disorders [39,40,41].

Sensitization (i.e., self-reported variables of sensitization and neuropathic symptoms) had an indirect effect on explaining the relationship between distress and severity in the current path model. This indirect mediating effect agrees with a recent meta-analysis reporting that psychological/emotional factors are associated with somatosensory function in people with musculoskeletal pain [42]. It seems clear that sensitization features have an exponential effect on distress, and vice versa; interaction promotes more severity. Nevertheless, it should be considered that the CSI is not proof of sensitization, but may act as a proxy for some of the manifestations present in people with chronic pain [43,44]. The clinimetric properties of the CSI are well-established, except for the content validity [43].

The model also found that sensitivity (i.e., PPTs) did not show a significant effect on the severity of the disease. Evidence supports the presence of widespread pressure hyperalgesia as an expression of sensitization in FMS [45]. Interestingly, the presence of sensitization is associated with poorer outcomes in response to conservative and surgical treatment in musculoskeletal pain conditions [46,47], explaining the heterogeneous response of patients with FMS to treatment. Nevertheless, it should be considered that PPT is a psychophysiological variable, i.e., a quantitative sensory test used for evaluating the patient’s response to a stimulus. In fact, linear associations between PPTs with pain and related disability outcomes are not generally reported in the literature [48]. Our path model supports this lack of association between sensitivity and severity. It is postulated that sensitivity (i.e., PPTs) reflects a neurophysiological mechanism (i.e., a mechanism construct), whereas severity (i.e., clinical features) reflects the clinical construct of pain, and a nonlinear influence can be expected. In fact, sensitization was associated with severity, which supports the possibility that self-reported associated symptoms are more relevant than objective measures, such as sensitivity (PPTs).

The International Association for the Study of Pain (IASP) considers FMS as “chronic primary pain” since it may be conceived as a disease in “its own right” [49]. Current and previous findings suggest that FMS can also be defined as a “nociplastic” pain condition associated with sensitization and psychological/cognitive disturbances [50]. The hypothesis that FMS is a nociplastic pain condition supports why exercise, a therapeutic strategy able to reduce sensitivity throughout adaptations in the central nervous system [51], exhibits the highest level of evidence for its treatment [52]. In fact, pain mechanisms underpinning each condition must be considered for optimizing the prescription of exercise for individuals with nociplastic pain such as FMS, supporting the complexity of these conditions [53].

Accordingly, current findings suggest that FMS management should include a multi-model program consisting of targeting pain mechanisms (e.g., pain education, physical therapy), emotional/psychological factors (e.g., cognitive behavior, coping strategies), physical condition (e.g., exercise), and behavioral change interventions focusing on aspects of a healthy lifestyle (e.g., sleep, stress, nutrition, smoking, and physical activity) [54]. These interventions should be individualized according to the predominant mechanism [55].

Although this study applied SEM to better understand the complexity of FMS, some limitations should be considered. First, the fact that this was a cross-sectional study precludes the ability to disentangle between–subjects from within–subjects relationships. For instance, we cannot distinguish whether distress is associated with severity because whenever people feel distressed it results in greater severity (a within-subjects effect) or because people who are on average distressed tend to exhibit greater severity (a between-subjects effect). Given that temporal precedence is a key requirement for determining causality, causal inference based on the current study should be made with caution. Second, in order to reduce the heterogeneous nature of the condition of interest and to account for gender differences in pain conditions, only women with FMS were included in this study. It is possible that the identified model would be different for men with FMS; this represents a relevant area for further research. Third, we tested widespread pressure pain sensitivity as a clinical feature of sensitization. It would be interesting to investigate other sensitization outcomes, e.g., thermal or electrical thresholds, conditioning pain modulation, or nociceptive flexor reflex, to determine different associations. Finally, we did not assess the personality features of our study’s subjects with FMS, which may be another factor to be considered regarding distress.

## 5. Conclusions

The proposed model suggests that distress and sensitization, mediated by poor sleep, exhibit complex mediating effects on severity in women with FMS. The role of sensitivity seems to be less relevant for severity. The identified path model can be leveraged in clinical trials by evaluating specific treatment approaches in FMS according to the predominant feature.

## Figures and Tables

**Figure 1 biomedicines-10-01188-f001:**
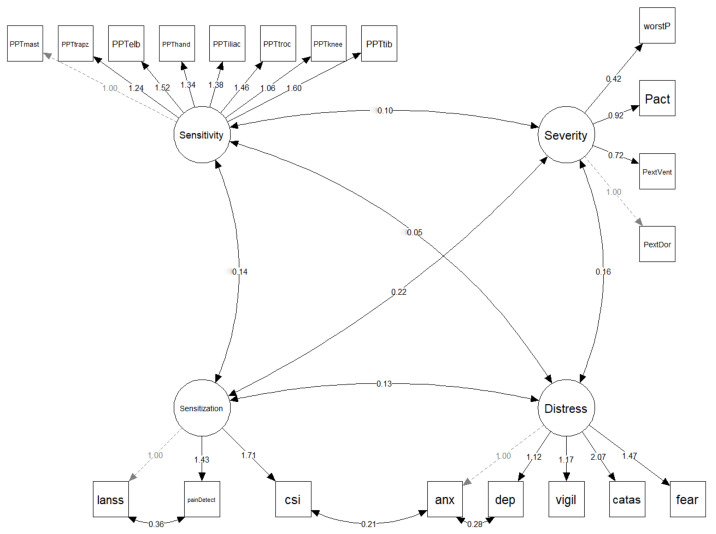
Theoretical latent variable model. Variables surrounded by a square box are observed variables, whilst those in a circle are latent variables. Numbers reflect standardized parameter estimates. Dotted arrows reflect fixed relationships. PPT: pressure pain threshold; mast: mastoid process; trapz: upper trapezius; elb: elbow; iliac: posterosuperior iliac spine; troc: greater trochanter; tib: tibialis anterior; worstP: worst pain at rest; Pact: pain during daily activities; PextVent: pain extent ventral surface; PextDor: pain extent dorsal surface; S-LANSS: self-reported Leeds Assessment of Neuropathic Symptoms and Signs; csi: Central Sensitization Inventory; anx: anxiety (HADS-A); dep: depression (HADS-D); vigil: Pain Vigilance and Awareness Questionnaire (PVAQ); catas: Pain Catastrophizing Scale (PCS); fear: Tampa Scale for Kinesiophobia (TSK-11).

**Figure 2 biomedicines-10-01188-f002:**
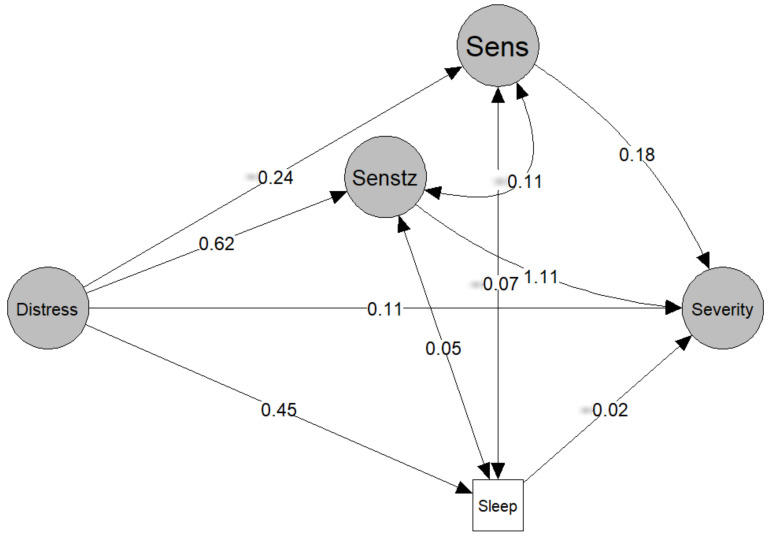
Structural equation model linking psychological distress to symptom severity in fibromyalgia. Values represent the standardized β coefficient linking the effect of the independent variable (tail of the arrow) to the dependent variable (head of the arrow). Senstz: sensitization; Sens: sensitivity. Observed variables are not plotted to reduce visual clutter.

**Table 1 biomedicines-10-01188-t001:** Descriptive characteristics of women with fibromyalgia syndrome (FMS, n = 113).

Variables	Summary Value
Age (years)	52.5(11)
Weight (kg)	71.4 (16.6)
Height (m)	1.60 (0.1)
BMI	27.5 (6.2)
Numbers of years diagnosed	10.2 (8.9)
Mean pain (NPRS, 0–10)	6.4 (1.7)
Worst pain (NPRS, 0–10)	7.3 (2.2)
Pain activity (NPRS, 0–10)	8.1 (1.9)
Pain extent ventral (%)	26.5 (24.7)
Pain extent dorsal (%)	31.3 (26.1)
PPT mastoid (kPa)	151.2 (90.8)
PPT trapezius (kPa)	125.5 (60.4)
PPT elbow (kPa)	149.0 (87.1)
PPT hand (kPa)	120.2 (59.1)
PPT posterior iliac (kPa)	233.9 (130.7)
PPT trochanter (kPa)	257.7 (123.9)
PPT knee (kPa)	148.1 (107.1)
PPT tibialis (kPa)	187.1 (108.7)
S-LANSS (0–24)	17.45 (5.45)
PainDETECT (0–38)	19.9 (7.1)
CSI (0–100)	70.7 (11.55)
HADS-A (0–21)	11.4 (3.7)
HADS-D (0–21)	10.0 (4.1)
Pain hypervigilance (PVAQ)	27.0 (8.2)
Catastrophizing (PCS, 0–52)	22.45 (12.25)
Kinesiophobia (Fear, TSK-11, 0–44)	24.9 (7.5)
Sleep quality (PSQI, 0–21)	13.8 (3.9)

BMI: Body Mass Index; NPRS: Numerical Pain Rate Scale; PPT: pressure pain thresholds; S-LANSS: self-reported version of the Leeds Assessment of Neuropathic Symptoms and Signs; CSI: Central Sensitization Inventory; HADS: Hospital Anxiety and Depression Scale (A: anxiety, D: depression); PCS: Pain Catastrophizing Scale; PVAQ: Pain Vigilance and Awareness Questionnaire; PSQI: Pittsburgh Sleep Quality Index; TSK-11: 11-item Tampa Scale for Kinesiophobia.

**Table 2 biomedicines-10-01188-t002:** Standardized parameter estimates for the model.

DV	IV	Coef	SE	LB	UB	Pval	Sig	Type
Indirect effect of sensitization		0.687	0.212	0.259	1.146	0.001	s	Med
Indirect effect of sensitivity		−0.042	0.048	−0.170	0.036	0.382	ns	Med
Indirect effect of sleep (PSQI)		−0.008	0.036	−0.095	0.069	0.826	ns	Med
Total		0.745	0.196	0.296	1.115	0.000	s	Med
Sensitivity	Distress	−0.239	0.123	−0.479	0.002	0.052	ns	Reg
Sensitization	Distress	0.618	0.191	0.244	0.993	0.001	s	Reg
Severity	Sensitization	1.113	0.317	0.496	1.734	0.000	s	Reg
Severity	Sensitivity	0.176	0.170	−0.157	0.512	0.300	ns	Reg
Severity	Sleep (PSQI)	−0.017	0.078	−0.170	0.136	0.825	ns	Reg
Severity	Distress	0.108	0.222	−0.327	0.543	0.627	ns	Reg
Sleep	Distress	0.452	0.209	0.041	0.863	0.031	s	Reg
Distress	Anx (HADS-A)	1.000	0.000					LV
Distress	Dep (HADS-D)	1.114	0.235	0.654	1.575	0.000	s	LV
Distress	Hypervigil (PVAQ)	1.170	0.274	0.636	1.706	0.000	s	LV
Distress	Catas (PCS)	2.069	0.421	1.248	2.898	0.000	s	LV
Distress	Fear (TSK-11)	1.463	0.331	0.814	2.113	0.000	s	LV
Sensitivity	PPTmast	1.000	0.000					LV
Sensitivity	PPTtrapz	1.237	0.224	0.798	1.676	0.000	s	LV
Sensitivity	PPTelb	1.518	0.145	1.232	1.803	0.000	s	LV
Sensitivity	PPThand	1.338	0.265	0.821	1.858	0.000	s	LV
Sensitivity	PPTiliac	1.380	0.134	1.117	1.641	0.000	s	LV
Sensitivity	PPTtroc	1.461	0.262	0.948	1.974	0.000	s	LV
Sensitivity	PPTknee	1.056	0.183	0.696	1.415	0.000	s	LV
Sensitivity	PPTtib	1.592	0.231	1.140	2.043	0.000	s	LV
Sensitization	S-LANSS	1.000	0.000					LV
Sensitization	PainDETECT	1.423	0.225	0.979	1.863	0.000	s	LV
Sensitization	CSI	1.705	0.434	0.855	2.554	0.000	s	LV
Severity	P_ext_Dor	1.000	0.000					LV
Severity	P_ext_Vent	0.723	0.212	0.308	1.139	0.001	s	LV
Severity	P_act	0.925	0.618	-0.284	2.143	0.134	ns	LV
Severity	Worst_P	0.423	0.338	-0.240	1.087	0.211	ns	LV

DV: dependent variable; IV: independent variable; Coef: coefficient; SE: standard error; LB: lower bound of 95% confidence interval; UB: upper bound of 95% confidence interval; Pval: *p* value; Sig: significant; s: statistically significant; ns: not statistically significant; Med: mediation; Reg: regression; LV: latent variable; PPT: pressure pain thresholds; S-LANSS: self-reported version of the Leeds Assessment of Neuropathic Symptoms and Signs; CSI: Central Sensitization Inventory; HADS: Hospital Anxiety and Depression Scale (A: anxiety, D: depression); PCS: Pain Catastrophizing Scale; PVAQ: Pain Vigilance and Awareness Questionnaire; PSQI: Pittsburgh Sleep Quality Index; TSK-11: 11-item Tampa Scale for Kinesiophobia; P_ext_Dor: pain extent dorsal; P_ext_Vent: pain extent ventral; P_act: pain during daily activities; Worst_P: worst pain.

## Data Availability

All data derived from this study are presented in the text.

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
