# Peer review of "Distress and Sensitization as Main Mediators of Severity in Women with Fibromyalgia: A Structural Equation Model"

_biomedicines, 2022, doi:10.3390/biomedicines10051188_

Round 1

Reviewer 1 Report

These authors intended to identify complex interactions between the observed and latent variables of FMS patients by means of a structural equation modelling (SEM) approach. They have succeeded in providing a new causal path model for this complex disorder. It should be mentioned in the text if this is the first attempt to apply the SEM approach to the symptoms of FMS or not.

Lines 246~247: They described that a significant indirect effect of Sensitization that explained the relationship between Distress and Severity, but it is not clearly written how they computed an indirect effect.

Table 2: DV (dependent variable), IV (independent variable), LB?, UB? should be explained in the Legend of the Table. What are Indirect 1~3, Indirect Contrast 1~3? The word "sleep" in the column of DV should be "Sleep".

Line 54: "important" should be "importance".

Author Response

Response Letter manuscript Biomedicines-1706617

Distress and Sensitization as Main Mediators of Severity in Women with Fibromyalgia: A Structural Equation Model

We would like to thank the reviewers for their comments, which have clarified some aspects of the manuscript. We have edited the text according to the suggestions from the reviewers. We have highlighted changes in yellow throughout the manuscript. A point-by-point response is presented below.

Reviewer 1

These authors intended to identify complex interactions between the observed and latent variables of FMS patients by means of a structural equation modelling (SEM) approach. They have succeeded in providing a new causal path model for this complex disorder. It should be mentioned in the text if this is the first attempt to apply the SEM approach to the symptoms of FMS or not.

Response: We would like to thank the reviewer for this positive feedback. To be honest, when we conducted the study, this wad the first attempt to apply SEM approaches in FMS. The reviewer 2 has advised us of a paper published on 3 May 2022 using SEM in FMS. After carefully reviewing that paper, we have included it in the introduction and discussion and still remark that our paper is the most complete SEM to date.

Lines 75-77: “A recent study has used SEM for assessing the relevance of psychological factors such as pain acceptance, catastrophizing, and coping strategies in pain severity in women with FMS [12]. However, this study only included psychological/cognitive, but not other type, of variables [12].”

Lines 269-271: “In fact, it has been recently observed, by using a SEM framework, the relevance of psychological/ cognitive factors in women with FMS [12].”

  1. Droppert KM, Knowles SR. he Role of pain acceptance, pain catastrophizing, and coping strategies: A Validation of the common sense model in females living with fibromyalgia. J Clin Psychol Med Settings. 2022. doi: 10.1007/s10880-022-09873-w.

Lines 246~247: They described that a significant indirect effect of Sensitization that explained the relationship between Distress and Severity, but it is not clearly written how they computed an indirect effect.

Response: In line 159 of the original manuscript (current line 163), we described how the indirect effect was calculated as seen below using the product of coefficient approach.

“In simple mediation models, with only one mediator, the effect of the independent variable on the mediator is denoted by 16a" style="width: 7pt; height: 14pt">, the effect of the mediator on the dependent variable is denoted by 16b" style="width: 7pt; height: 14pt">, the direct effect of the independent variable on the dependent variable is denoted by 16c'" style="width: 10pt; height: 14pt">, the indirect effect of the mediator by the product 16a√ób" style="width: 28pt; height: 14pt">, or simply 16ab" style="width: 13pt; height: 14pt">. In a parallel mediation model with three mediators, there are three 16a" style="width: 7pt; height: 14pt">, 16b" style="width: 7pt; height: 14pt">, and 16ab" style="width: 13pt; height: 14pt">.

Table 2: DV (dependent variable), IV (independent variable), LB?, UB? should be explained in the Legend of the Table. What are Indirect 1~3, Indirect Contrast 1~3? The word "sleep" in the column of DV should be "Sleep".

Response: We have ensured all abbreviations are added to the table legends, thanks for let us know. The indirect 1-3 meant “Indirect effect of”, which we have reworded. The Indirect contrast was a typo and meant to represent the difference between two indirect pathways. Since we did not report the difference indirect pathways, we have removed this from the table as we agree with the reviewer, it was confusing.

Line 54: "important" should be "importance".

Response: We have changed the word.

We hope that the current version of the paper can be finally accepted for publication in BIOMEDICINES

Sincerely yours,

The authors

Reviewer 2 Report

Dear Authors, I think that it is a very interesting paper. Please explain better the meaning of the numbers in Fig. 1 and 2, and how we can know if the relationship is strong, fear....

Besides, recently was published a similar paper by Droppert et al (2022), can you discuss your results at the light also of those?

Author Response

Response Letter manuscript Biomedicines-1706617

Distress and Sensitization as Main Mediators of Severity in Women with Fibromyalgia: A Structural Equation Model

We would like to thank the reviewers for their comments, which have clarified some aspects of the manuscript. We have edited the text according to the suggestions from the reviewers. We have highlighted changes in yellow throughout the manuscript. A point-by-point response is presented below.

Reviewer 2

Dear Authors, I think that it is a very interesting paper. Please explain better the meaning of the numbers in Fig. 1 and 2, and how we can know if the relationship is strong, fear....

Response: We would like to thank the Reviewer for this positive feedback. We have added to the Figure captions the requested information. Numbers reflect the standardised regression or correlation estimates.

In line 216, we have defined how to interpret the parameters on the standardized scale. Given that all continuous variables were standardized to a mean of zero and standard deviation of 1, it enables comparison of different pathways on a common scale. The bigger the magnitude the stronger the relationship. Standardized correlation estimates are not classified as strong, mid or moderate as other simple correlation coefficients. We believe that with the definition included in this version, readers will understand the magnitude.

Besides, recently was published a similar paper by Droppert et al (2022), can you discuss your results at the light also of those?

Response: We would like to thank to the reviewer for this paper, since we are unaware of it. We have now included it in the introduction and discussion as follows:

Lines 75-77: “A recent study has used SEM for assessing the relevance of psychological factors such as pain acceptance, catastrophizing, and coping strategies in pain severity in women with FMS [12]. However, this study only included psychological/cognitive, but not other type, of variables [12].”

Lines 269-271: “In fact, it has been recently observed, by using a SEM framework, the relevance of psychological/ cognitive factors in women with FMS [12].”

  1. Droppert KM, Knowles SR. he Role of pain acceptance, pain catastrophizing, and coping strategies: A Validation of the common sense model in females living with fibromyalgia. J Clin Psychol Med Settings. 2022. doi: 10.1007/s10880-022-09873-w.

We hope that the current version of the paper can be finally accepted for publication in BIOMEDICINES

Sincerely yours,

The authors